# Proteo-genomic analyses in relatively lean Chinese adults identify proteins and pathways that affect general and central adiposity levels

Andri Iona[1,7], Pang Yao[1,7], Alfred Pozarickij[1], Christiana Kartsonaki[1], Saredo Said[1], Neil Wright [1], Kuang Lin[1], Iona Millwood [1], Hannah Fry [1], Mohsen Mazidi[1], Baihan Wang[1], Yiping Chen [1], Huaidong Du [1], Ling Yang [1], Daniel Avery [1], Dan Schmidt[1], Dianjianyi Sun [2,3,4], Pei Pei[2], Jun Lv[2,3,4], Canqing Yu [2,3,4], Michael Hill [1], Junshi Chen[5], Fiona Bragg [1,6], Derrick Bennett [1], Robin Walters [1], Liming Li [2,3,4], Robert Clarke [1], Zhengming Chen [1] ✉ & On behalf of China Kadoorie Biobank Collaborative Group*

Adiposity is an established risk factor for multiple diseases, but the causal relationships of different adiposity types with circulating protein biomarkers have not been systematically investigated. We examine the causal associations of general and central adiposity with 2923 plasma proteins among 3977 Chinese adults (mean BMI = 23.9 kg/m²). Genetically-predicted body mass index (BMI), body fat percentage (BF%), waist circumference (WC), and waist-to-hip ratio (WHR) are significantly (FDR < 0.05) associated with 399, 239, 436, and 283 proteins, respectively, with 80 proteins associated with all four and 275 with only one adiposity trait. WHR is associated with the most proteins ($n = 90$) after adjusting for other adiposity traits. These associations are largely replicated in Europeans (mean BMI = 27.4 kg/m²). Two-sample Mendelian randomisation (MR) analyses in East Asians using *cis*-protein quantitative trait locus (*cis*-pQTLs) identified in GWAS find 30/2 proteins significantly affect levels of BMI/WC, respectively, with 10 showing evidence of colocalisation, and seven (inter-alpha-trypsin inhibitor heavy chain H3, complement factor B, EGF-containing fibulin-like extracellular matrix protein 1, thioredoxin domain-containing protein 15, alpha-2-antiplasmin, fibronectin, mimecan) are replicated in separate MR using different *cis*-pQTLs identified in Europeans. These findings identified potential novel mechanisms and targets, to our knowledge, for improved treatment and prevention of obesity and associated diseases.

The prevalence of overweight and obesity continues to increase steadily in most countries, including China, with excess weight now affecting 2 billion adults worldwide[1]. However, the distribution of body fat differs substantially by sex and ancestry worldwide[2,3]. For a given weight, East Asian populations have greater central adiposity than Western populations[3]. Likewise, men also have greater levels of visceral adipose tissue than women[2]. These ancestry differences in the distribution of body fat distribution may account for different risks of associated disease or mechanisms through which adiposity increases the risk of such diseases.

Adiposity is an established risk factor for several major diseases, including cardio-metabolic disease, cancer and infection, but the

[1]Clinical Trial Service Unit, Nuffield Department of Population Health, University of Oxford, Oxford, UK. [2]Peking University Center for Public Health and Epidemic Preparedness and Response, Beijing, China. [3]Department of Epidemiology and Biostatistics, School of Public Health, Peking University, Beijing, China. [4]Key Laboratory of Epidemiology of Major Diseases, Ministry of Education, Peking University, Beijing, China. [5]China National Center for Food Safety Risk Assessment, Beijing, China. [6]Health Data Research UK Oxford, University of Oxford, Oxford, UK. [7]These authors contributed equally: Andri Iona, Pang Yao. *A list of authors and their affiliations appears at the end of the paper. ✉e-mail: zhengming.chen@ndph.ox.ac.uk

mechanisms may vary for individual diseases[4,5]. It is well-established that the effects of adiposity for cardiometabolic diseases are chiefly mediated by type 2 diabetes (T2D)[5], and dyslipidaemia[6–10] and blood pressure[11,12]. For instance, adipose tissue also releases adipokines, including adiponectin and resistin, and these proteins may influence insulin sensitivity and risk of T2D[13]. Adipose tissue also produces pro-inflammatory cytokines, including tumour necrosis factor alpha (TNF-alpha) and interleukin-6 (IL-6), which promote inflammation and interfere with insulin signalling, leading to insulin resistance and increased risk of T2D[14]. Hence, many of these protein-related biomarkers are interconnected with each other and may influence levels of each other and risks of different diseases.

Previous studies of adiposity and protein biomarkers have been constrained by a limited number of biomarkers, and use of single (usually body mass index [BMI]) rather than multiple measures of adiposity. Moreover, there is a general lack of genetic data to assess the causal relevance (e.g. through Mendelian randomisation [MR] and colocalisation analyses) and direction (i.e. adiposity-to-protein or protein-to-adiposity) of these associations[15–22]. Furthermore, previous studies have been limited to Western populations where most adults are overweight or obese[16–20,22]. In China, despite a recent increase in the prevalence of obesity[1], the mean levels of adiposity among adults remain substantially lower than those in Western populations[3]. A comprehensive investigation of the associations of general and central adiposity with levels of circulating proteins in Chinese adults could have potential for identification of novel pathways through which different types of adiposity affect multiple disease risks.

We conducted detailed observational and genetic analyses of four adiposity traits with almost 3000 circulating proteins in 3977 Chinese adults selected from the China Kadoorie Biobank (CKB). The study aimed to (i) identify plasma proteins significantly associated with general (BMI, body fat percentage [BF%]) and central (waist circumference [WC], waist-hip ratio [WHR]) adiposity; (ii) clarify the direction, strength and causal relevance of the observed associations; (iii) explore the possible biological mechanisms through which different adiposity traits may increase disease risks; (iv) assess whether certain associated proteins can causally affect levels of adiposity; and (v) compare and replicate the associations in a European-ancestry population using the UK Biobank study.

## Results

Among the 3977 participants, the mean baseline age was 57.3 (standard deviation [SD] 11.6) years, and the mean BMI was 23.9 (3.3) kg/m$^2$, mean BF% 25.4 (6.3), mean WC 81.9 (9.1) cm, and mean WHR 0.89 (0.001; Table 1). Overall, 6% of participants were obese (i.e. BMI ≥ 30 kg/m$^2$), slightly higher in women and in urban residents. BMI and WC were positively associated with blood pressure and inversely associated with smoking and physical activity (Table 1 and Supplementary Table 1). These associations were broadly similar in Ishaemic heart disease (IHD) cases and sub-cohort participants, although IHD cases had higher mean levels of blood pressure than sub-cohort participants (Supplementary Table 2). BMI, BF%, WC and WHR were intercorrelated ($r > 0.6$; Supplementary Fig. 1).

### Observational associations of different adiposity measures with proteins

Overall, BMI, BF%, WC and WHR were significantly associated at false discovery rate (FDR) < 0.05 with plasma levels of 1809 (Batch 1/Batch 2: 1088/721), 1838 (1115/723), 1808 (1099/709), and 1736 (1055/681) proteins, respectively (Supplementary Table 3, Figs. 1 and 2). There were 1513 proteins significantly associated with all four adiposity measures and 61, 89, 12, and 68 proteins uniquely associated with BMI, BF%, WC and WHR, respectively (Supplementary Fig. 2a). Among the overlapping proteins, the associations with BF% were slightly stronger than those with other adiposity measures (Supplementary Fig. 3). The three proteins showing the strongest positive associations with all four adiposity measures were leptin, fatty acid-binding protein, adipocyte (FABP4), and scavenger receptor cysteine-rich domain-containing group B protein (SSC4D). Insulin-like growth factor-binding protein 2 (IGFBP2) and insulin-like growth factor-binding protein

1 (IGFBP1) consistently showed the strongest inverse associations with all four adiposity measures, while the top three most strongly inversely associated proteins were serum paraoxonase/lactonase 3 (PON3) for BMI and WC, collagen alpha-1(IV) chain (COL4A1) for BF% and lipoprotein lipase (LPL) for WHR. The associations of all four adiposity measures with all individual proteins are shown in Supplementary Data 1. In multivariable regression analyses with adjustment for each other, BMI, BF%, WC, and WHR were independently associated at FDR < 0.05 with levels of 339, 1018, 26, and 620 proteins, respectively (Supplementary Fig. 4).

After applying the Bonferroni significance threshold, 1259 (Batch 1/Batch 2: 798/461), 1312 (836/476), 1303 (824/479), and 1242 (782/460) proteins remained significantly associated with BMI, BF%, WC and WHR, respectively (Supplementary Table 4). Analyses of these significantly associated proteins indicated that 1089 proteins were associated with both general and central adiposity measures (e.g. proto-oncogene tyrosine-protein kinase receptor Ret [RET], FURIN), and 30, 56, 14 and 40 were uniquely associated with BMI (e.g. apolipoprotein M [APOM], coiled-coil domain-containing protein 80 [CCDC80]), BF% (e.g. growth/differentiation factor 15 [GDF15], erythropoietin [EPO]), WC (e.g. natural cytotoxicity triggering receptor 1 [NCR1], junctional adhesion molecule B [JAM2]) and WHR (e.g. neural cell adhesion molecule L1 [L1CAM], sialic acid-binding Ig-like lectin 9 [SIGLEC9]), respectively (Supplementary Fig. 2b).

In sex-specific analyses, slightly more proteins were associated with adiposity among women than men (BMI: 1547/1453, 1234 overlapped; BF%: 1593/1490, 1274 overlapped; WC: 1570/1484, 1270 overlapped; WHR: 1513/1395, 1183 overlapped at FDR < 0.05). Among these overlapping proteins in men and women for specific adiposity traits, the strength of the associations were similar in men and women ($r > 0.97$; Supplementary Fig. 5).

In sensitivity analyses restricted to subcohort participants only, BMI, BF%, WC and WHR were associated at FDR < 0.05 with 1619, 1669, 1575 and 1491 proteins, respectively (Supplementary Fig. 6), with ~97% overlapping overall and all directionally consistent with results from the main analyses. Furthermore, the results remained largely consistent following additional adjustments for physical activity, smoking, and alcohol, with ~99% of the associations consistent with those observed in the main analyses (Supplementary Table 5).

### Genetic associations of different adiposity measures with proteins

In genetic analyses, a total of 662 proteins were significantly associated at FDR < 0.05 with at least one of the four genetically-predicted adiposity traits, including 80 proteins associated with all adiposity traits (Fig. 2 and Supplementary Table 3). Among these 80 proteins, the associations were all directionally consistent ($r > 0.86$), but were stronger with measures of central than general adiposity (Supplementary Fig. 3). There were 275 proteins associated with only one of the four adiposity traits (BMI: 79; BF%: 22; WC: 76; WHR: 98; Supplementary Fig. 2a). Overall, >90% of these genetic associations were also significant in observational analyses (Fig. 3). In genetic analyses, the three proteins showing the strongest positive associations with BMI, %BF and WC were FABP4, leptin, and a disintegrin and metalloproteinase with thrombospondin motifs 15 (ADAMTS15) while for WHR they were protein disulfide-isomerase (P4HB), beta-glucuronidase (GUSB) and soluble scavenger receptor cysteine-rich domain-containing protein (SSC5D). The protein showing the strongest inverse associations with BMI and WC was PON3, while for BF% and WHR they were NT-3 growth factor receptor (NTRK3) and IGFBP2, respectively. In multivariable MR analyses, BMI, BF%, WC, and WHR were each independently associated with levels of 9 (lymphocyte-specific protein 1 [LSP1], lntegrin beta-2 [ITGB2], protein GOLM2 [GOLM2], secretogranin-1 [CHGB], myelin-oligodendrocyte glycoprotein [MOG], desmoglein-4 [DSG4], retbindin [RTBDN], amyloid-like protein 1 [APLP1], and phosphoprotein associated with glycosphingolipid-enriched microdomains 1 [PAG1]), 2 (leptin and FABP4), 1 (cadherin-15 [CDH15]), and 90 (e.g. P4HB, SSC5D, low-density lipoprotein receptor [LDLR], inactive serine protease PAMR1 [PAMR1],

**Table 1 | Baseline characteristics of participants by quintiles of measured BMI**

| Characteristics[a] | Quintiles of BMI, kg/m$^2$ | | | | | All ($n$ = 3977) |
|---|---|---|---|---|---|---|
| | Q1 ($n$ = 796) | Q2 ($n$ = 789) | Q3 ($n$ = 812) | Q4 ($n$ = 785) | Q5 ($n$ = 795) | |
| Age, sex and socioeconomic factors | | | | | | |
| Age, years (SD) | 58.8 (15.1) | 56.2 (12.5) | 56.3 (11.9) | 56.9 (12.1) | 56.9 (12.8) | 57.3 (11.6) |
| Women, % | 52.9 | 51.2 | 53.0 | 51.0 | 62.6 | 53.7 |
| Urban, % | 28.1 | 39.2 | 53.0 | 59.6 | 65.6 | 48.8 |
| ≥6 years of education, % | 43.4 | 45.8 | 46.7 | 46.4 | 44.4 | 45.1 |
| Anthropometry, blood pressure and fasting time, mean (SD) | | | | | | |
| BMI, kg/m$^2$ | 19.2 (1.3) | 21.9 (0.6) | 23.7 (0.5) | 25.7 (0.6) | 29.2 (1.9) | 23.9 (3.3) |
| Body fat percentage | 18.1 (3.1) | 22.4 (3.2) | 25.2 (3.4) | 28.3 (3.8) | 33.2 (4.9) | 25.4 (6.3) |
| Waist circumference, cm | 70.5 (5.1) | 77.2 (5.5) | 81.6 (5.4) | 86.6 (5.1) | 94.1 (9.5) | 81.9 (9.1) |
| Waist-to-hip ratio | 0.84 (0.002) | 0.87 (0.002) | 0.89 (0.002) | 0.91 (0.002) | 0.95 (0.002) | 0.89 (0.001) |
| SBP, mmHg | 129.8 (21.8) | 133.6 (19.3) | 138.8 (22.0) | 142.1 (21.9) | 146.5 (23.7) | 138.3 (22.0) |
| Fasting time, hours | 4.4 (4.9) | 4.7 (4.3) | 4.9 (4.1) | 4.7 (3.8) | 4.3 (3.5) | 4.7 (4.1) |
| Lifestyle factors | | | | | | |
| Ever regular smoker, % | | | | | | |
| Men | 82.8 | 75.4 | 73.2 | 70.0 | 75.8 | 75.0 |
| Women | 5.6 | 7.4 | 6.2 | 4.0 | 5.1 | 5.8 |
| Regular alcohol drinker, % | | | | | | |
| Men | 34.6 | 35.4 | 36.4 | 36.0 | 31.0 | 34.6 |
| Women | 3.2 | 2.5 | 2.0 | 4.1 | 3.6 | 3.0 |
| Physical activity, MET-h/day (SD) | 17.6 (11.2) | 17.4 (10.5) | 17.2 (12.1) | 17.4 (10.3) | 16.0 (10.0) | 17.3 (10.7) |
| Health status and medical history,[b] % | | | | | | |
| Self-rated poor health | 13.2 | 16.5 | 16.6 | 18.7 | 18.0 | 16.6 |
| Diabetes | 5.9 | 7.9 | 11.1 | 13.3 | 17.7 | 11.2 |
| Chronic kidney disease | 1.0 | 1.0 | 1.6 | 1.7 | 1.5 | 1.4 |
| Cancer | 0.8 | 0.8 | 0.4 | 0.4 | 0.9 | 0.6 |

[a]Adjusted for age, sex and study area, as appropriate.
[b]Based on self-report, while for diabetes, screen-detected diabetes at baseline was also included.
*SD* standard deviation, *BMI* body mass index, *SBP* systolic blood pressure, *MET* metabolic equivalent of task.

coagulation factor IX [F9]) proteins, respectively (Supplementary Fig. 4). After applying the Bonferroni correction threshold, a total of 193 proteins were significantly associated with any of the four adiposity traits, with 4 proteins associated with all four traits and 116 with a single trait (BMI: 33; BF%: 2; WC: 50 WHR: 31). Supplementary Data 1 shows the MR results for all individual proteins.

In sensitivity analyses restricted to subcohort participants only, 94, 114, 259, 230 proteins were significantly associated with BMI, BF%, WC and WHR, respectively (Supplementary Fig. 6). Over 95% of these proteins overlapped with the overall results and all the associations were directionally consistent. Analyses using East Asian-specific BMI genetic score (GS) replicated ~75% of the associations found in main analyses using trans-ancestry GS, which were directionally consistent (Supplementary Fig. 7). The MR-Egger intercept test showed evidence of directional pleiotropy for one protein (6-phosphogluconate dehydrogenase, decarboxylating [PGD]), and the MR-PRESSO global test showed no evidence for horizontal pleiotropy at FDR < 0.05 (Supplementary Data 2).

**Analyses in the UK Biobank**
In UK Biobank participants (mean BMI: 27.4 kg/m$^2$), a total of 2529 (Batch 1/Batch 2: 1375/1154), 2525 (1364/1161), 2524 (1367/1157) and 2471 (1342/1129) proteins were significantly (FDR < 0.05) associated with BMI, BF%, WC and WHR, respectively (Supplementary Fig. 8), which replicated >95% of the significant associations in CKB conventional analyses. In genetic analyses, 1455, 1596, 1647 and 1711 proteins showed significant associations with genetically-derived BMI, BF%, WC

and WHR, respectively, which replicated ~92% of the associations observed in CKB genetic analyses (Fig. 2). Moreover, there were high correlations ($r$ > 0.85) between the effect sizes for the overlapping proteins in CKB and UK Biobank.

**Enrichment analyses**
Of the 193 proteins significantly associated (Bonferroni < 0.05) with adiposity in genetic analyses, three proteins (LPL, PON3 and gamma-synuclein [SNCG]) were also associated with T2D in a two-sample MR (2SMR) analyses using CKB cis-protein quantitative trait locus (*cis*-pQTLs) in Asian Genetic Epidemiology Network (AGEN). In enrichment analyses, there was strong evidence of Gene Ontology (GO) enrichment for these 193 proteins in biological processes related to atherosclerosis, lipid metabolism, regulation of blood pressure, inflammation, immune function, and other biological processes (Fig. 4). For 86 BMI-related proteins, the most enriched biological processes were regulation of chemotaxis, leukocyte proliferation, and regulation of leukocyte proliferation. For BF%-related proteins, the most enriched processes included regulation of blood pressure; for WC-related proteins it was cytokine-mediated signalling pathway; for the 53 WHR-related proteins, they were lipid regulation (including cholesterol or sterol homeostasis), lipid localization, regulation of lipid localization and lipid transport. For WHR, 10 out of the 53 proteins were associated with lipids. Three of these 10 proteins were uniquely associated with central adiposity (phosphatidylcholine-sterol acyltransferase [LCAT], fibroblast growth factor receptor 4 [FGFR4] and Adiponectin [ADIPOQ]). Further details of all significantly enriched biological processes for the proteins

## a) Adiposity-to-Protein

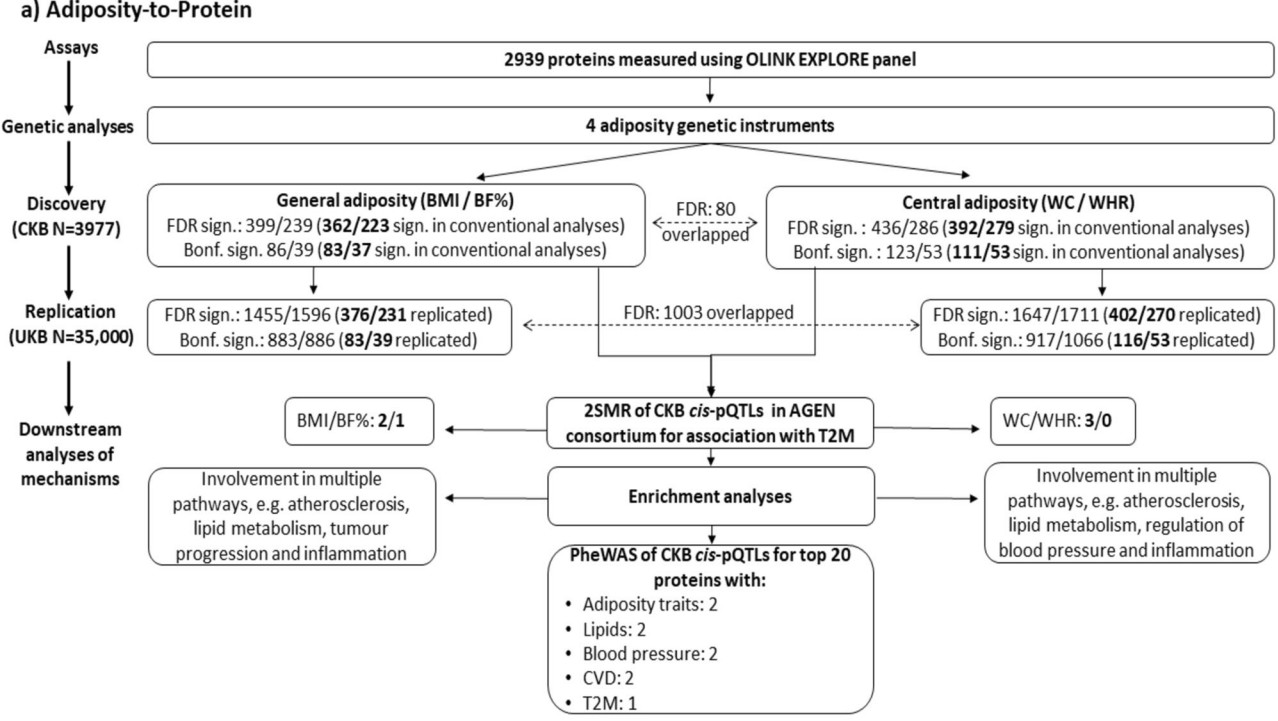

## b) Protein-to-Adiposity

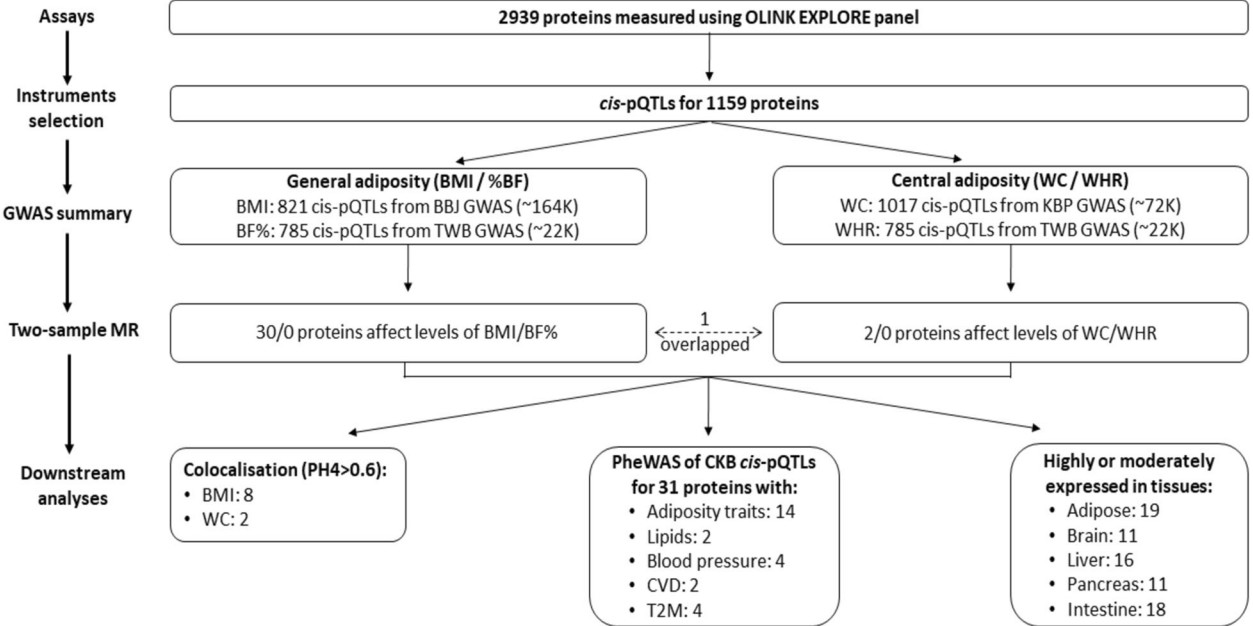

**Fig. 1 | Overview of study design, analytic approaches and key findings.** Figures (**a**) and (**b**) represent the overview of study design, analytic approaches and key findings from adiposity to protein direction (**a**) and from protein to adiposity direction (**b**), respectively. 2SMR two-sample Mendelian randomization, AGEN Asian Genetic Epidemiology Network, BBJ Biobank Japan, BF% body fat percentage, BMI body mass index, Bonf. Bonferroni, *cis*-pQTLs cis-protein quantitative trait locus, CKB China Kadoorie Biobank, CVD cardiovascular disease, FDR false discovery rate, GWAS genome-wide association study, KBP Korean Biobank Project, sign. significant, T2D type 2 diabetes, TWB Taiwan Biobank, UKB UK Biobank, WC waist circumference, WHR waist-to-hip ratio.

associated with the four adiposity traits are shown in Supplementary Data 3. Of these 193 proteins, *cis*-pQTLs were identified for 109 proteins in CKB genome-wide association study (GWAS), which were used for phenome-wide association study (PheWAS) analyses (Supplementary Data 4). For the top 20 selected proteins (i.e. top 5 for each adiposity measure), *cis*-pQTLs for 2 proteins (agrin [AGRN], calcitonin [CALCA]) were associated, based on

PhenoScanner, with several adiposity traits. In addition, *cis*-pQTLs for FURIN were associated with lipids, blood pressure, and cardiovascular disease (CVD; Fig. 1 and Supplementary Data 4). After accounting for the potential pre-enrichment of proteins included in the Olink Explore panels, no terms were enriched for BMI, one enriched for BF% (negative regulation of multicellular organismal process), one enriched for WC (response to

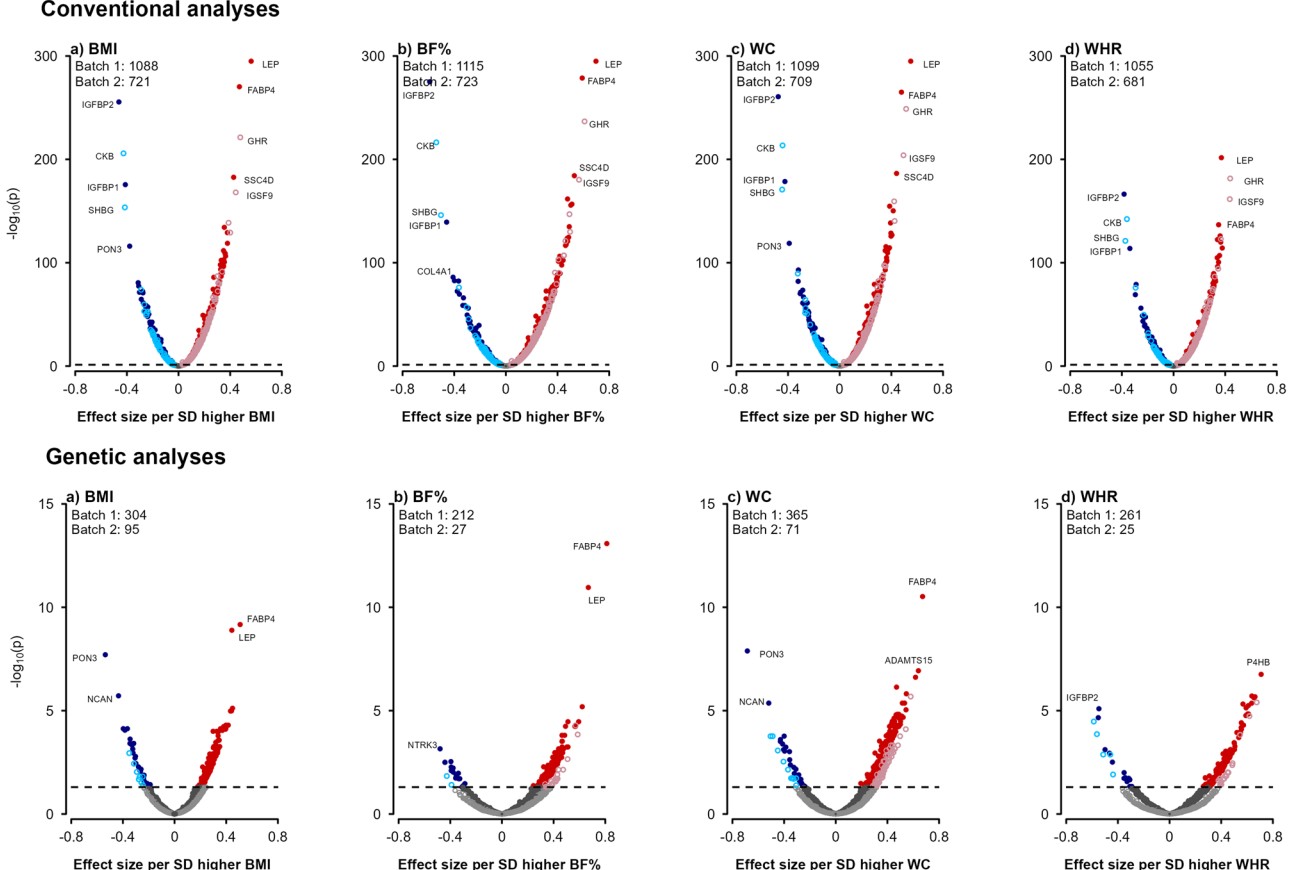

**Fig. 2 | Associations of protein biomarkers with 1-SD higher BMI, BF%, WC and WHR in conventional and genetic analyses.** Figures (**a**–**d**) represent the associations of BMI, BF%, WC and WHR, respectively, with protein biomarkers. The x-axis represents the effect size of the association between the adiposity traits and the protein biomarkers, while the y-axis indicates the –log10 p-value. Red dots denote positive FDR corrected associations, blue dots denote negative FDR corrected associations, and grey dots denote non-significant associations. Solid dots denote Olink batch 1, while open dots denote batch 2. Analyses were adjusted for age, age[2], sex, study area, fasting time, ambient temperature, plate ID, ascertainment status, and the first 11 PCs (for genetic analyses only). BF% body fat percentage, BMI body mass index, SD standard deviation, WC waist circumference, WHR waist-to-hip ratio.

abiotic stimulus) and three lipid metabolism-related terms enriched for WHR (Supplementary Table 6).

### Genetic analyses of the effects of proteins on adiposity

In CKB, *cis*-pQTLs for 1159 out of the 2939 proteins were identified in GWAS, which were then used in 2SMR analyses to investigate the effects of protein levels on adiposity (Fig. 1). In a 2SMR analyses of Biobank Japan (BBJ) involving 821 *cis*-pQTLs, 30 proteins (e.g. all-trans-retinol dehydrogenase [ADH1B], succinate-semialdehyde dehydrogenase, mitochondrial [ALDH5A1], BMP-binding endothelial regulator protein [BMPER], secretory carrier-associated membrane protein 3 [SCAMP3], and inter-alpha-trypsin inhibitor heavy chain H4 [ITIH4]) significantly affected levels of BMI after correction for multiple testing (Supplementary Table 7), including two (BMPER and SCAMP3) that showed bi-directional associations. Moreover, in a 2SMR analysis of Korean Biobank Project involving 1017 *cis*-pQTLs, two proteins (aldehyde dehydrogenase, mitochondrial [ALDH2], ITIH4) significantly affected levels of WC. There was correlation between these 31 proteins that affected levels of BMI and/or WC (Supplementary Fig. 9).

In colocalisation analyses, there was evidence (posterior probability H4 > 0.6; Supplementary Table 8) of shared causal genetic variants for 11 of these 31 proteins with BMI (9 proteins) and WC (2 proteins; Supplementary Fig. 10). After applying more stringent threshold of H4 > 0.8, 3 (ADH1B, Mimecan [OGN], alpha-2-antiplasmin [SERPINF2]) and 2 (ALDH2, ITIH4) proteins shared strong evidence of colocalisation with BMI and WC, respectively. Independent two-sample MR analyses involving (i) CKB and

the Taiwan Biobank did not provide substantial evidence of proteins significantly impacting levels of BF% and WHR; (ii) UK Biobank *cis*-pQTLs and Genetic Investigation of Anthropometric Traits (GIANT) GWAS summary statistics replicated seven (inter-alpha-trypsin inhibitor heavy chain H3 [ITIH3], complement factor B [CFB], EGF-containing fibulin-like extracellular matrix protein 1 [EFEMP1], thioredoxin domain-containing protein 15 [TXNDC15], SERPINF2, fibronectin [FN1], OGN) and one (ITIH3) of the 30 proteins that affected levels of BMI using East Asian *cis*-pQTLs and summary statistics, when using GIANT including and not including UK Biobank participants (Supplementary Table 7). Among these 10 proteins (posterior probability H4 > 0.6), independent 2SMR analyses, using CKB *cis*-pQTLs and published GWAS summary statistics, showed evidence of causal associations with stroke (ALDH2), IHD (SERPINF2), T2D (natural cytotoxicity triggering receptor 3 ligand 1 [NCR3LG1], ALDH2, ITIH4, apolipoprotein E [APOE], ADH1B), low-density lipoprotein cholesterol (LDL-C; APOE, ALDH2, SERPINF2, FN1, ITIH4) and systolic blood pressure (SBP; FN1).

Of the 31 proteins that affected levels of BMI and/or WC, a PheWAS analysis using PhenoScanner showed that the *cis*-pQTLs for 14 proteins (e.g. ADH1B, apolipoprotein B receptor [APOBR], APOE) were linked with several adiposity-related traits, including BMI, WC, and body composition (Supplementary Table 9). Furthermore, *cis*-pQTLs for two proteins (APOE and complement factor B [CFB]) were associated with lipids, while *cis*-pQTLs for four proteins were associated with systolic blood pressure (ADH1B, APOE, CFB, and coagulation factor XIII B chain [F13B]). Notably, APOE and FN1 were associated with coronary artery disease. In

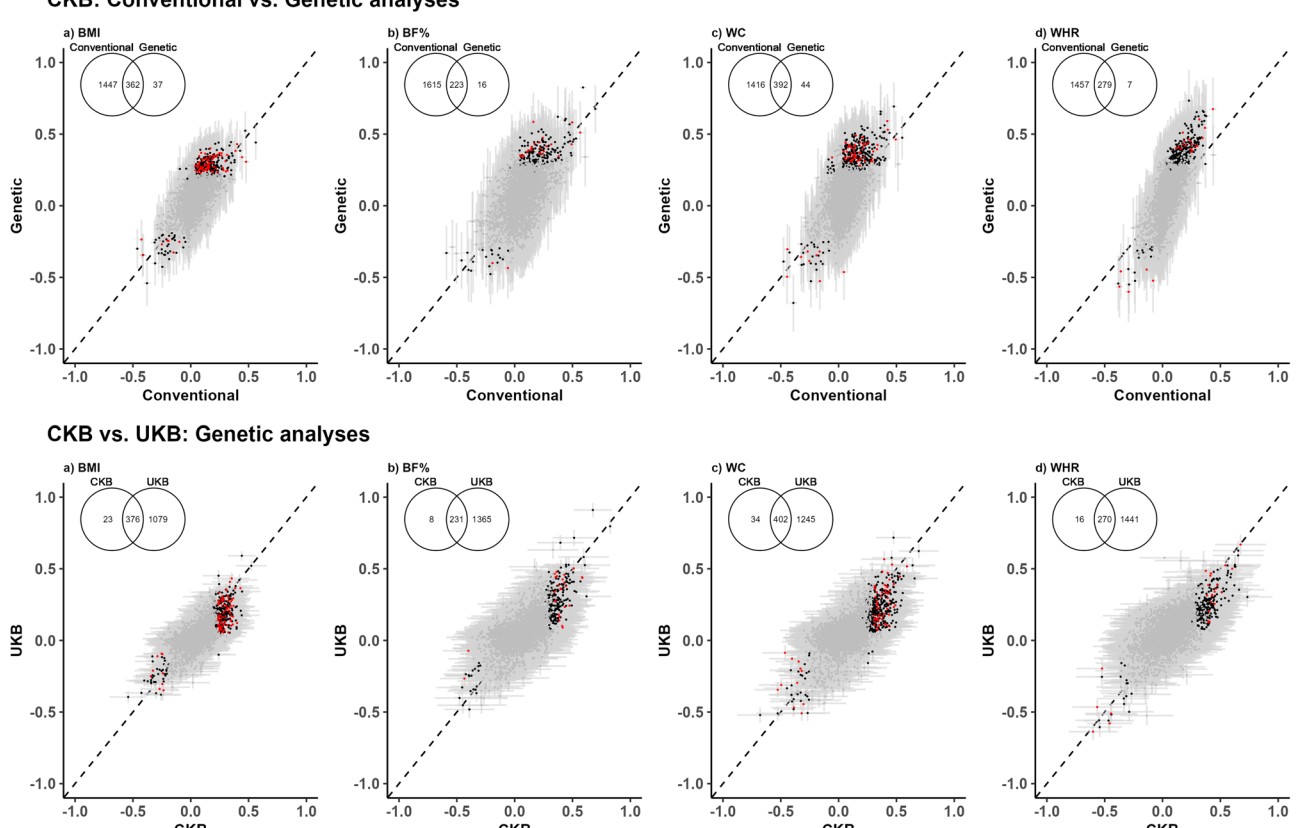

**Fig. 3 | Comparison of associations of protein biomarkers with 1-SD higher BMI, BF%, WC and WHR in conventional vs. genetic analyses in CKB and CKB vs. UKB genetic analyses.** Figures (**a–d**) represent the comparison of the associations of BMI, BF%, WC, and WHR, respectively, with protein biomarkers, either between CKB conventional and genetic analyses or between CKB and UKB genetic analyses. The x-axis and y-axis represent the effect size of the associations between the adiposity traits and the protein biomarkers. Black dots denote Olink batch 1 FDR-corrected associations, red dots denote batch 2 FDR-corrected associations, while grey dots denote non-significant associations, and grey lines indicate the 95% confidence intervals. In CKB, analyses were adjusted for age, age$^2$, sex, study area, fasting time, ambient temperature, ascertainment status, and the first 11 PCs (for genetic analyses only). In UKB, analyses were adjusted for age, age$^2$, sex, assessment centre, fasting time, plate ID, and the first 40 PCs (for genetic analyses only). BF% body fat percentage, BMI body mass index, CKB China Kadoorie Biobank, SD standard deviation, UKB UK Biobank, WC waist circumference, WHR waist-to-hip ratio.

tissue-specific expression analyses, six proteins were found to be highly expressed in adipose tissue (e.g. ADH1B, APOE, ALDH2), seven in liver (e.g. CFB, complement factor H-related protein 2 [CFHR2], SERPINF2) and four in brain (peceptor-type tyrosine-protein phosphatase zeta [PTPRZ1], APOE, ALDH5A1 and NCR3LG1). All other proteins were expressed in multiple tissues (Supplementary Table 7; Supplementary Fig. 11). A further search of Open-Target provided evidence supporting drug targets for six proteins (ADH1B, ALDH5A1, APOE, CFB, FN1, ALDH2), with three having approved drugs that target conditions such as age-related macular degeneration (ADH1B), eye disease (FN1), parasitic infection or treatment of alcohol addiction (ALDH2), and bipolar disorder or epilepsy (ALDH5A1; Supplementary Table 7).

## Discussion

This is the first study to comprehensively examine the associations of general and central adiposity with plasma levels of 2923 proteins. We found that different adiposity traits were likely causally associated with >650 proteins in Chinese adults, with >90% of the associations replicated in Europeans who had higher levels of adiposity. Using *cis*-pQTLs from CKB GWAS, we identified 31 proteins that significantly affected levels of adiposity. Furthermore, colocalisation analyses demonstrated that 10 of these 31 proteins had shared causal genetic variants with adiposity, providing further support, along with findings in PheWAS and enrichment analyses, of their causal relevance and potential as possible targets for drug development for prevention and treatment of obesity and obesity-related diseases.

While several previous observational studies have investigated associations of adiposity, chiefly BMI, with plasma levels of proteins[15–21], these studies typically involved populations with high mean levels of adiposity and varying numbers of proteins measured by different platforms, but few such studies assessed likely causality and direction of the associations with adiposity traits[15–21]. In an observational study of 4600 participants from Germany, the UK and Qatar involving 921 SomaScan proteins, 152 (16.5%) proteins were significantly associated with BMI[20]. In another UK-based study of 2737 participants involving 3622 SomaScan proteins, self-reported and genetically-derived BMI was significantly associated with 1576 (43.5%) and eight (e.g. leptin, FABP4, paired immunoglobulin-like type 2 receptor alpha [PILRA] and inhibin beta B chain [INHBB]) proteins, respectively[18]. In a recent report of ~50,000 UK Biobank participants involving 2923 Olink proteins, 2348 proteins were significantly associated (after Bonferroni correction) with BMI, but the latter study did not include any genetic analyses[22]. The present study extended our previous analyses of BMI with 1463 proteins in CKB[21] to include four separate adiposity traits and 2923 Olink proteins, with replication in the UK Biobank for both observational and genetic findings. We showed that although there were substantial overlaps in the number of specific proteins associated with different adiposity traits, WC was associated with more proteins compared with other three adiposity traits in genetic analyses. Moreover, the strength of the genetic associations were somewhat greater for central (i.e. WC and WHR) than for general adiposity for the overlapping proteins. Enrichment analyses of selected top proteins showed multiple pathways through which adiposity may increase

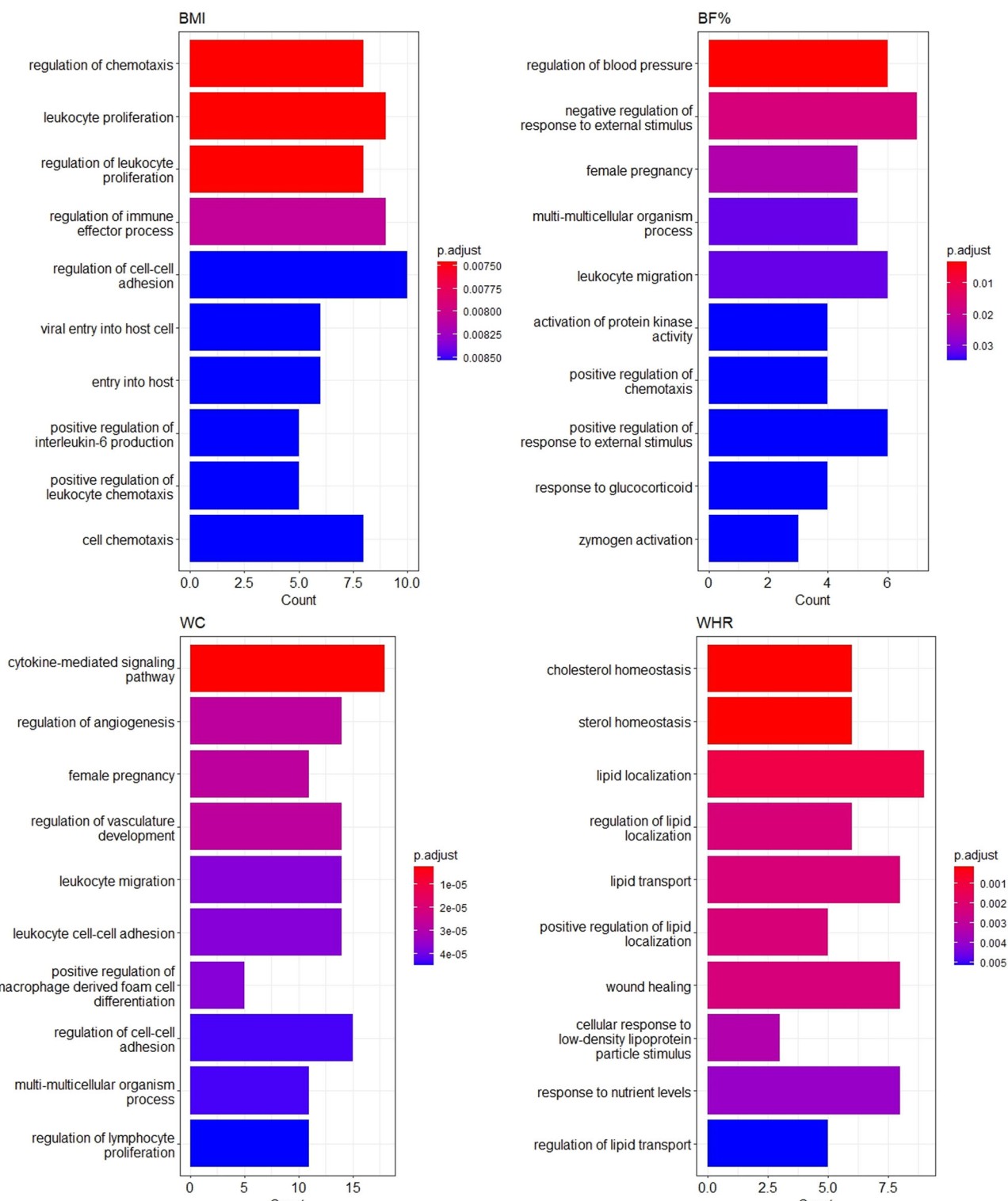

**Fig. 4 | Enriched GO biological terms for proteins casually affected by different adiposity traits.** The bar charts represent the top 10 Gene Ontology (GO) terms for each adiposity trait, showcasing the number of proteins significantly associated with each term. The GO terms are ordered based on the significance level of their associations. BF% body fat percentage, BMI body mass index, WC waist circumference, WHR waist-to-hip ratio.

risks of specific diseases, with certain pathways being more strongly linked to specific adiposity traits (e.g. WHR and lipid metabolism). Our multi-variable MR analyses revealed likely distinct impacts of BMI, BF%, WC, and WHR on protein levels. For example, BMI influenced 9 proteins, including LSP1, ITGB2, and CHGB, implicating inflammation[23], cell adhesion[24], and neuroendocrine functions, while BF% affected levels of leptin and FABP4, both crucial in fat metabolism and energy homeostasis[25]. On the other hand, WC was only associated with CDH15, which is linked to muscle structure[26], but WHR appeared to show the broadest effect, affecting levels of 90 proteins, such as P4HB, SSC5D, and LDLR, which were involved in protein folding, cardiac immune response[27], and lipid metabolism[28]. Notably, proteins such as IGFBP2, p-selectin (SELP), and LPL were also affected by

WHR, highlighting the involvement of insulin-like growth factor regulation[29], cell adhesion[30], and lipid transport[31], respectively. As measures of central obesity, it is likely that WHR are more strongly related to visceral adipose tissue, which is more metabolically active, more strongly related to insulin resistance[13,32], and hence more strongly associated with certain protein biomarkers and risk of cardio-metabolic diseases risk. Using protein *cis*-pQTLs identified in CKB GWAS, the 2SMR analyses found that 30 proteins significantly affected levels of BMI and two affected levels of WC, including one (ITIH4) that also affected BMI, while for BF% and WHR there were too few overlapping *cis*-pQTLs in publically available GWAS summary statistics to yield meaningful discovery. Two proteins (BMPER and SCAMP3) demonstrated bi-directional associations with BMI and the specific mechanisms for such associations are not clear and might involve possible feedback mechanisms, shared genetic pathways, or partially confounded by other adiposity traits. Indeed, for these two proteins, the BMI-to-protein associations were not significant after mutual adjustment for other adiposity measures, suggesting that the initial associations were likely mediated through shared pathways with other adiposity measures, rather than being direct effects. In addition, among the 31 proteins showing causal relevance for adiposity levels, only 1 protein (CFB) is in the MHC region, which is known for its high genetic complexity and extensive linkage disequilibrium and can potentially confound the association with adiposity. In separate colocalisation analyses we found evidence of shared causal genetic variants of 10 proteins with BMI (ten) and/or WC (two), which, together with PheWAS findings on adiposity traits, provided strong evidence for the causal relevance of these proteins for adiposity. Furthermore, 9 of these 10 proteins were causally related with cardio-metabolic diseases or their risk factors (e.g. LDL-C, SBP), suggesting that these proteins may impact on disease risks partly through adiposity. Indeed, our mediation MR analysis indicated that ~30% of the total effect of ALDH2 on T2D is mediated through WC (indirect effect *p*-value < 0.001). These findings align with previous research demonstrating the role of ALDH2 in metabolic health[33], with ALDH2 mutation (reduced activity) in mice showing increased susceptibility to obesity, glucose intolerance, and fatty liver due to impaired energy expenditure and adaptive thermogenesis. Of these 10 proteins, two (ADH1B and APOE) were highly expressed in adipose tissue and four (ADH1B, APOE, ITIH4, and SERPINF2) in the liver, which could be prioritised as promising candidates for drug treatment of obesity and associated diseases. ADH1B is a gene that encodes alcohol dehydrogenase 1B, an enzyme primarily involved in alcohol metabolism in the liver. ADH1B also regulates FABP4 expression during subcutaneous adipocyte differentiation and is inversely correlated with obesity and insulin resistance[34]. Thus, approved drug targeting ADH1B for age-related macular degeneration could be repurposed for treatment of obesity. SERPINF2, also known as alpha-2-antiplasmin, inhibits plasmin and regulates blood clotting through fibrinolysis[35]. Although the pathways through which SERPINF2 may affect adiposity are not fully understood, it is possible that it may include fibrosis of adipose tissue which has been linked to insulin resistance[36]. The causal relevance of SERPINF2 with BMI demonstrated in both Chinese and UK adults, in addition to the evidence of a causal association with IHD and LDL-C observed in the present study, provide strong support for SERPINF2 as a likely novel target, to our knowledge, for treatment of obesity and associated diseases. Another protein known to affect adiposity levels is GDF15, along with its receptors (GDNF family receptor alpha-like [GFRAL] and RET)[37]. Our genetic analyses using CKB *cis*-pQTLs and East-Asian summary statistics for adiposity showed that GDF15 and RET significantly affected WC, but not levels of other adiposity traits.

The present study had several strengths including the large numbers of proteins analysed, availability of four different measures of adiposity, use of strong trans-ancestry adiposity genetic instruments, and independent replication of the main results in different ancestry populations with typically different adipose tissue distribution. Moreover, we investigated the genetic associations using both MR and colocalisation analyses, along with assessment of direction of the associations (i.e. adiposity-to-protein and

protein-to-adiposity) and a range of downstream analyses to clarify multiple biological processes and the potential of selected individual proteins for future drug development. However, the present study also had several limitations. First, the genetic analyses lacked adequate statistical power to explore any apparent sex differences in the biological mechanisms related to adiposity. Second, our two-sample MR analyses only involved a very small number of proteins due to lack of overlapping *cis*-pQTLs in publically available GWAS summary statistics for East Asian populations, particularly for BF%, WC and WHR. Consequently, we were unable to confirm (or refute) previous findings suggesting certain proteins (e.g. leptin, advanced glycosylation end product-specific receptor [AGER], dermatopontin [DPT], and CTSA) that may affect adiposity, nor were we able to identify potential novel associations of proteins with specific adiposity traits. Third, our genetic analyses did not involve *trans*-pQTLs, which, while subject to pleiotropy effects, may offer new insights. Future studies involving a larger sample size, diverse populations, imaging-based measures of adiposity, and improved genetic instruments, involving perhaps multiple *cis*-pQTLs and *trans*-pQTLs, are needed to clarify the effects of different proteins, including their interactions, on general and central adiposity (or vice versa).

In summary, the present study of relatively lean Chinese adults demonstrated that adiposity was significantly associated with a large number of proteins, with genetic evidence for the causal relevance of >650 proteins in the adiposity-to-protein direction, with many being unique to specific adiposity traits. Moreover, we identified >30 proteins that causally affect adiposity levels. These findings, together with enrichment analyses and available experimental data of multiple pathways, provide support for novel protein targets, to best of our knowledge, for potential treatment of obesity and obesity-related diseases.

## Methods

### Study population
Detailed information about the CKB design and methods have been reported previously[38,39]. CKB recruited 512,869 participants aged 30–79 years from 10 (5 urban, 5 rural) geographically diverse areas across China during 2004–2008. At the initial baseline survey, invited participants attended study clinics in local communities and completed an interviewer-administered laptop-based questionnaire (on sociodemographic and lifestyle factors, and medical history and medication), physical measurements (e.g. blood pressure, heart rate, height and weight, bioelectrical impedance analysis, waist and hip circumferences), and provided a 10 mL non-fasting blood sample (with time since last meal recorded) for random plasma glucose testing and long-term storage. Prior international, national and regional ethical approvals were obtained and all participants provided written informed consent in the study. All ethical regulations relevant to human research participants were followed.

### Anthropometric measurements
Anthropometric measurements were recorded with participants wearing light clothing without shoes. Standing height was measured to the nearest 0.1 cm using a portable stadiometer and weight was measured using a body composition analyser (TANITA-TBF-300GS; Tanita Inc., Tokyo, Japan), with subtraction of the weight of clothing (typically 0.5 kg in summer, 1.0 kg in spring/autumn and 2.0–2.5 kg in winter). WC and hip circumference (HC) were measured to the nearest 0.1 cm, using a non-stretchable tape, over underclothing or, if over clothes, with 1.0–2.0 cm and 1.0–2.5 cm (typically, 1.0 cm if measured over a skirt and 2.5 cm if over trousers), respectively, subtracted from the measurements. WC was measured midway between the lowest rib and the iliac crest and HC was measured around the maximum circumference of the buttocks. BF% was derived from bioelectrical impedance analysis, which was measured using the foot-to-foot TANITA-TBF-300GS body composition analyser. BMI was calculated as weight in kilograms divided by the square of the height in meters (kg/m$^2$). WHR was calculated as the ratio of WC to HC.

## Study design and proteomics assay

The present study involved 3977 participants from a case-cohort study within CKB (1951 incident IHD cases and a randomly selected sub-cohort of 2026 participants; Supplementary Fig. 12)[21]. The participants had no prior history of CVD, no history of lipid-lowering drug (e.g. statins) use at the time of sample collection, but had genome-wide genotyping data available.

Stored baseline plasma samples from these participants were retrieved, thawed, and aliquoted into 96-well plates (with 8 wells per plate reserved for quality control [QC] assays), with each plate containing both cases and controls in varying proportions, plated in the order that they were retrieved from storage at the Wolfson laboratory in Oxford. To minimize inter- and intra-run variation, the samples were randomised across plates and normalized using both an internal control and an inter-plate control and then transformed using a pre-determined correction factor[40]. Plasma levels of 2923 proteins were measured using the Olink Explore 3072 panel in two separate batches. The first batch (1463 unique proteins) was assayed in the Olink laboratory in Uppsala, Sweden and the second batch (1460 unique proteins) was assayed in the Olink laboratory in Boston, USA, with each batch covering all samples.

The limit of detection (LOD) was determined using negative control samples (buffer without antigen). Sample were flagged as having QC warning if the incubation control deviated more than a pre-determined value (±0.3) from the median value of all samples on the plate (but values below the LOD were included in the analyses). Individual samples were flagged for assay warnings if values deviated by ≥3-fold from negative controls. Details of individual proteins, their distributions and the number of QC or sample warnings per assay, in addition to proportions with values below LOD, are shown in Supplementary Data 5, Supplementary Fig. 13 and Supplementary Table 10. The pre-processed data were provided in the arbitrary normalized protein eXpression unit on a log2 scale.

## Genotyping and genetic instruments for general and central adiposity

Overall ~100,000 CKB participants were genotyped using a custom-designed 800K- single nucleotide polymorphism (SNP) array (Axiom [Affymetrix]), which included ~76,000 participants randomly selected from the overall cohort, from whom the sub-cohort of 2026 individuals was selected[41]. Detailed information about the GWAS and development of genetic instruments have been reported previously[42]. In summary, GWAS were conducted for four adiposity traits in both the CKB and UK Biobank, using rank-inverse normal-transformed data and a sex-stratified approach. The Han and Eskin random effects (HE-RE) model was used to meta-analyse results to better detect associations, accounting for effect size heterogeneity. Subsequent analyses were limited to variants with a minor allele frequency (MAF) ≥ 0.01, applying a significance threshold of $P < 5 \times 10^{-8}$. In UK Biobank, 400 to 675 loci were associated with each trait, while in CKB, 21 to 71 loci met the significance criteria. The variants were merged into 480 to 789 conditionally independent associations in UK Biobank, and 22 to 73 in CKB.

Trans-ancestry meta-analysis was conducted for each adiposity trait by combining sex-stratified GWAS data from both UK Biobank and CKB. The HE-RE model was employed to accommodate effect size heterogeneity across ancestries. The trans-ancestry GSs for BMI, BF%, WC and WHR were derived using loci associated at genome-wide significance in sex-combined trans-ancestry GWAS in CKB and UK Biobank, which involved a total of 816, 692, 604 and 439 variants, respectively, with MAF ≥0.01, and were weighted according to their effect sizes in UK Biobank. These GS (F-statistic: 149, 78, 100, 64, respectively) explained 3.7%, 1.9%, 2.5% and 1.5% of variance, respectively (Supplementary Table 11), and were not associated with confounders such as smoking or alcohol drinking (Supplementary Table 12). In sensitivity analysis we also used East Asian-specific BMI GS developed using dosages of 73 bi-allelic SNPs, with MAF ≥0.01 that were extracted from CKB GWAS of BMI (variance explained=1.1%).

## Discovery of pQTL

Detailed information about the GWAS for pQTLs discovery has been reported previously[43]. Briefly, a GWAS of 2939 proteins among 3974 CKB participants identified 2091 autosomal pQTLs at genome-wide significance ($P \leq 5 \times 10^{-8}$). Of the 1863 proteins with at least one pQTL, 1159 had cis-pQTLs, accounting for 37.3% of all pQTLs identified.

## Statistical analysis

The baseline characteristics of study participants are reported by quintiles of BMI, with the prevalence or mean values of selected variables standardised to the age (5-year groups), sex and study area structure of the cases and sub-cohort. Plasma protein levels were standardised (i.e. values divided by their SD) and analysed as continuous variables. Linear regression was used to examine the associations of different measures of adiposity (i.e. BMI, BF%, WC and WHR) with protein biomarkers, adjusting for age, age[2], sex, study area, fasting time, ambient temperature, plate ID and case-subcohort ascertainment. For each protein, we further estimated the adjusted differences and 95% CIs per 1-SD higher levels of specific adiposity measures. We also assessed the conditionally independent effects of the adiposity traits on protein levels using a multivariable regression model, in which protein levels were regressed on BMI, BF%, WC, and WHR simultaneously. In sensitivity analyses, additional adjustments were made for physical activity, smoking, and alcohol, both individually and in combination.

For MR analyses, the 2-stage least squares estimation method was used to relate genetically-instrumented BMI, BF%, WC and WHR with proteins. First, the associations between the adiposity GS and corresponding adiposity measures were examined using linear regression, adjusting for age, age[2], sex, study area, fasting time, ambient temperature, case-subcohort ascertainment and the first 11 national principal components. Second, associations of the resulting predicted adiposity values with specific proteins were examined using linear regression employing the same adjustments (including plate ID) except principal components. We calculated the genetically-instrumented associations of per 1-SD higher levels of each adiposity measure (i.e. BMI: 3.6 kg/m²; BF%: 6.3; WC: 9.1 cm; WHR: 0.001) with measured proteins levels, to permit comparisons with each other and with the observational analyses. The robustness of MR results was tested using MR and MR-PRESSO R packages, employing inverse-variance weighted MR, MR-Egger, MR-PRESSO, and weighted median MR methods to address potential pleiotropic effects and ensure reliable causal estimates. For significant proteins in MR analyses (adiposity-to-protein direction) after Bonferroni correction, we conducted GO and Kyoto Encyclopedia of Genes and Genomes (KEGG) enrichment analyses using clusterProfiler (v.4.2.2)[44], to determine which biological functions or processes were significantly enriched based on hypergeometric tests. In sensitivity analyses, we further compared all 2923 Olink proteins to annotated proteins to account for potential pre-enrichment of proteins in the Olink explore panels.

We also undertook separate observational and genetic analyses in UK Biobank to replicate the main study findings, examining the same 2923 Olink proteins in ~35,000 randomly selected participants of the UK Biobank Pharma Proteomics Project, with exclusion of participants with prior-CVD or use of cholesterol-lowering medication[19]. The conditionally independent effects of the adiposity traits on protein levels were also assessed using multivariable MR.

To assess whether some individual proteins could causally affect adiposity levels, we undertook separate 2SMR analyses using protein cis-pQTLs obtained from CKB GWAS[43] and GWAS summary statistics of adiposity traits including from BBJ[45] (BMI), Korean Biobank Project[46] (WC), and Taiwan Biobank[47] (BF% and WHR). Colocalisation analyses were performed to assess whether adiposity and proteins have a shared signal at cis-pQTL. Firstly, fine-mapping was performed using the Sum of Single Effects model (SuSiE; susieR v0.12.16 R package)[48] to identify 95% credible sets of likely causal variants. For each of cis-pQTLs, SuSiE was ran using an internal CKB LD reference and specifying the maximum number of non-zero effects in the SuSiE regression to 10, the minimum

absolute correlation allowed in a credible set to 0.1, and the maximum number of iterations to 100,000. When the number of credible sets was found to be equal to ten fine-mapping, a larger number of non-zero effects in the SuSiE regression model was used. Colocalization using SuSiE output was performed using coloc (v5.2.1) package in R. In addition, 2SMR analyses using *cis*-pQTLs obtained from GWAS of UK Biobank, and summary statistics of BMI from GIANT with ($n = \sim700,000$) and without ($n = \sim210,000$) UK Biobank participants were conducted to replicate any such associations[49,50]. Both 2SMR analyses used the Wald ratio method[51].

We used PhenoScanner (v2)[52,53] to investigate associations of *cis*-pQTLs from CKB with a range of phenotypes using a *P* value threshold of $5 \times 10^{-8}$. For certain proteins that causally affected the levels of adiposity in our analyses, we also assessed whether they were also causally associated with other cardio-metabolic traits and diseases, such as T2D[54], SBP[55], LDL-C[56], IHD[57,58] and stroke[59], using 2SMR analyses and protein *cis*-pQTLs obtained from CKB GWAS and published GWAS summary statistics. In addition, we screened the protein expression database of Genotype-Tissue Expression (GTEx; v8)[60] to examine the tissue-specific role of the causal proteins in obesity.

All statistical analyses were performed using R version 4.1.2 and packages 'Epi', 'tidyverse', 'ggplot2', 'ckbplotr', 'MendelianRandomization' and 'MRPRESSO'. To correct for multiple testing within each Olink batch, we used Benjamini-Hochberg FDR or the more stringent Bonferroni-corrected thresholds for defining statistical significance.

## Reporting summary
Further information on research design is available in the Nature Portfolio Reporting Summary linked to this article.

## Data availability
Data from baseline, first and second resurveys, and disease follow-up are available under the CKB Open Access Data Policy to bona fide researchers. Sharing of genotyping data is constrained by the Administrative Regulations on Human Genetic Resources of the People's Republic of China. Access to these and certain other data is available through collaboration with CKB researchers. Details of the CKB Data Sharing Policy are available at www.ckbiobank.org. A research proposal will be requested to ensure that any analysis is performed by bona fide researchers. Researchers who are interested in obtaining additional information or data that underlines this paper (*CKB Research Track No.: 2022-0036)* should contact ckbaccess@ndph.ox.ac.uk. For any data that is not currently available to open access, researchers may need to develop formal collaboration with study group.

## Code availability
The statistical analyses in this manuscript were conducted using R version 4.1.2 and several published R packages, including 'Epi,' 'tidyverse,' 'ggplot2,' 'ckbplotr,' 'MendelianRandomization,' and 'MRPRESSO.'

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

## Acknowledgements

The chief acknowledgment is to the participants, the project staff, and the China CDC and its regional offices for assisting with the fieldwork. The members of the China Kadoorie Biobank collaborative group are listed in the Supplementary Information. We thank Judith Mackay in Hong Kong; Yu Wang, Gonghuan Yang, Zhengfu Qiang, Lin Feng, Maigeng Zhou, Wenhua Zhao, and Yan Zhang in China CDC; Lingzhi Kong, Xiucheng Yu, and Kun Li in the Chinese Ministry of Health; and Sarah Clark, Martin Radley, and Mike Hill in the CTSU, Oxford, for assisting with the planning, conduct and organization of the study. The CKB baseline survey and the first re-survey were supported by the Kadoorie Charitable Foundation in Hong Kong. The long-term follow-up and subsequent resurveys have been supported by Wellcome grants to Oxford University (212946/Z/18/Z, 202922/Z/16/Z, 104085/Z/14/Z, 088158/Z/09/Z) and grants from the National Natural Science Foundation of China (82192901, 82192903, 82192904, 82192900) and from the National Key Research and Development Program of China (2016YFC0900500). The UK Medical Research Council (MC_UU_00017/1, MC_UU_12026/2, MC_U137686851), Cancer Research UK (C16077/A29186, C500/A16896) and the British Heart Foundation (CH/1996001/9454), provide core funding to the Clinical Trial Service Unit and Epidemiological Studies Unit at Oxford University for the project. The proteomic assays were supported by BHF (18/23/33512), Novo Nordisk and Olink. DNA extraction and genotyping were supported by GlaxoSmithKline and the UK Medical Research Council (MC-PC-13049, MC-PC-14135). The trans-ancestry adiposity genetic scores were developed using data from UK Biobank (Application No.: 50474).

## Author contributions

A.I., P.Y., Z.C. contributed to the concept and design of the study. P.Y. and A.I. conducted statistical analyses and drafted the manuscript. A.P. conducted colocalisation analyses. A.I., P.Y., A.P., C.K., S.S., N.W., K.L., I.M., M.M., B.W., Y.C., H.D., F.B., D.B., J.L., C.Y., J.C., R.W., L.L., R.C.I., and Z.C. were involved in the planning, acquisition and interpretation of data.

I.M., H.F., H.D., Y.C., D.A., D.S., P.P., and M.H. provided administrative, technical, or material support. All authors provided critical revision of the manuscript for important intellectual content. A.I., P.Y., and Z.C. are the guarantors of this work and take responsibility for the integrity and accuracy of the data analysis. Z.C. supervised the work.

## Competing interests

The authors declare no competing interests.

## Ethics

The China Kadoorie Biobank (CKB) complies with all the required ethical standards for medical research on human subjects. Ethical approvals were granted and have been maintained by the relevant institutional ethical research committees in the UK and China.

## Informed consent

All participants provided written informed consent.

## Additional information

## China Kadoorie Biobank Collaborative Group

**Andri Iona**[1,7], **Pang Yao**[1,7], **Alfred Pozarickij**[1], **Christiana Kartsonaki**[1], **Saredo Said**[1], **Neil Wright**[1], **Kuang Lin**[1], **Iona Millwood**[1], **Hannah Fry**[1], **Mohsen Mazidi**[1], **Baihan Wang**[1], **Yiping Chen**[1], **Huaidong Du**[1], **Ling Yang**[1], **Daniel Avery**[1], **Dan Schmidt**[1], **Dianjianyi Sun**[2,3,4], **Pei Pei**[2], **Jun Lv**[2,3,4], **Canqing Yu**[2,3,4], **Junshi Chen**[5], **Fiona Bragg**[1,7], **Derrick Bennett**[1], **Robin Walters**[1], **Liming Li**[2,3,4], **Robert Clarke**[1] & **Zhengming Chen**[1] ✉

