## [Peer Review File · Communications Biology]

Reviewers' comments:

Reviewer #1 (Remarks to the Author):

The authors examined the observational and genetic analyses of four adiposity traits with 2923 plasma proteins among 3977 Chinese adults. They found that different adiposity traits were likely causally associated with >650 proteins in Chinese adults, with >90% of the associations replicated in Europeans who had higher levels of adiposity. Using cis-pQTLs in Chinese, they identified 31 proteins that significantly affected levels of adiposity. Overall, this is a well-conducted Proteo-genomic analyses. Specific comments are listed below.

1.The authors used piecewise non-linear MR analyses to assess the shape of the associations between adiposity measures and proteins. However, there is a substantial bias in nonlinear MR estimates for specifically BMI. Therefore, I have great concerns about the validity of this part of the manuscript.

2.It is not clear why the authors did not adjust for physical activity when they examined the associations of adiposity measures with protein biomarkers.

3.To evaluate the causal effects of proteins on adiposity measures, the authors performed two-sample MR using GWAS summary statistics from Asian population. However, the GSs of adiposity measures were derived using IVs obtained from the trans-ancestry GWAS. I suggest adding sensitivity analyses using IVs from Asian population only.

4.Two proteins (BMPER and SCAMP3) showed bi-directional associations. Does this mean that the genetically predicted levels of BMPER and SCAMP3 have effects on adiposity measures and the genetic predicted adiposity measures also have effects on the expression of BMPER and SCAMP3? How to explain such phenomeno?

Minor comments:

1.Line 162 “The GSs of for BMI”

Reviewer #2 (Remarks to the Author):

The authors investigated the associations between adiposity and 2,923 plasma proteins in 3,977 Chinese adults and further utilized bidirectional two-sample Mendelian randomization (MR) analysis to evaluate whether certain proteins influence adiposity. Through comprehensive analysis, the authors uncovered a series of relationships between proteins and obesity. However, I have some comments:

1. The authors did not provide detailed information on how the genetic scores for body mass index, body fat percentage, waist circumference and Waist-to-hip ratio were constructed. Detailed descriptions would help readers better understand the methodology and evaluate the validity of the genetic scores used in the analysis.
2. In the Protein-to-Adiposity section, the authors did not describe the screening process for cis-pQTLs in detail. The corresponding pQTL information or complete GWAS data should be provided.
3. In Figure 1a, the number of FDR sign proteins in the discovery set does not match the results section and eTable 7 of the manuscript. Please clarify the discrepancy.
4. In the colocalization analysis, the authors defined a posterior probability of $H_4 > 0.5$ as "strong evidence." However, it is more common to define a posterior probability of $H_4 > 0.8$ as "strong evidence." Please justify the chosen threshold.
5. In lines 425-428, the authors stated that these proteins with causal relationships with adiposity also have causal relationships with some cardio-metabolic diseases or their risk factors. They then directly inferred that adiposity might mediate the relationship between the proteins and cardiovascular metabolic diseases. This conclusion is not sufficiently rigorous. The authors should supplement the appropriate mediation analysis to support this viewpoint.
6. I appreciate the authors' use of four indicators to describe adiposity from different perspectives and their classification of general adiposity and central adiposity. Unfortunately, the authors did not compare and discuss the proteins associated with these two different types of adiposity in the discussion section.

Reviewer #3 (Remarks to the Author):

In this work, Iona et al. performed observational and Mendelian randomization (MR) analyses to investigate the associations between Olink-measured circulating protein levels and four adiposity traits. The authors identified significant associations and explored potential underlying biological mechanisms. The results were compared to those obtained based on the UK Biobank European ancestry population. This work can be an important data resource for proteogenomic studies, which enrich the representation of East Asian populations in literature. However, I have several concerns regarding the current study design and methodology.

1. It is unclear to me why the authors performed adiposity-to-protein association testing using genetic scores. Genetic scores for adiposity traits can be subject to high horizontal pleiotropy as many instruments are pleiotropic. What efforts have the authors made to

address risk of horizontal pleiotropy?

2. Following point #1, why didn't the authors use large-scale East Asian-specific GWAS for adiposity traits to perform formal MR analyses? The authors used genetic scores with instruments' coefficients "weighted according to their effect sizes in UKB" - This can lead to bias due to population specificity. Overall, the adiposity-to-protein "causal relationships" are over-stated.

3. The four adiposity traits are strongly correlated with each other - I suggest adding multivariable MR (and perhaps multivariable regression in observational analyses as well) to quantify joint/conditionally independent effects of these traits on protein levels.

4. The study focused on consensus, but for associations that were different between Chinese and European population, do they represent population-specific biological mechanisms? I believe such associations are more interesting than those that could be replicated - isn't this the point of having resources for ancestrally diverse populations?

5. In protein-to-adiposity MR analyses, again, how did the authors address risk of horizontal pleiotropy? Can a cis-pQTL be eQTL or sQTL of another gene that does not code for the assayed protein? In general, the MR analyses in the current form are overly simplistic and have few sensitivity analyses. The authors should consider accompanying the analyses with the STROBE-MR checklist.

6. For colocalized associations, please add locus-zoom plots for visual inspection.

7. Some of the proteins showing significant associations seem to be in the MHC region. These should not be the focus of interpretation due to exceedingly high risk of pleiotropy and complicated LD structure.

8. Non-linear MR, especially the adopted residual stratification method, has been criticized. See [https://www.thelancet.com/journals/landia/article/PIIS2213-8587\(23\)00364-9/fulltext](https://www.thelancet.com/journals/landia/article/PIIS2213-8587(23)00364-9/fulltext); The revised doubly-ranked stratification method has also been shown to introduce bias. See <https://www.medrxiv.org/content/10.1101/2023.08.21.23293658v1>. I suggest removing the non-linear MR analyses.

9. The enrichment test did not seem to account for the background. Are the Olink-measured ~3,000 proteins already enriched in cardiometabolic pathways?

10. Will the pQTL summary statistics be openly available?

Response to reviewers' comments

(Manuscript COMMSBIO-24-0892-T)

We have revised the manuscript in light of helpful comments from the three reviewers. The reviewers' verbatim comments are shown below in bold (but any purely background descriptive comments have been omitted), and our responses are shown as non-bold text. Line numbers indicating where changes have been made refer to the "tracked" changes version of the manuscript.

REVIEWERS' COMMENTS

REVIEWER #1

1. The authors used piecewise non-linear MR analyses to assess the shape of the associations between adiposity measures and proteins. However, there is a substantial bias in nonlinear MR estimates for specifically BMI. Therefore, I have great concerns about the validity of this part of the manuscript.

Response: We share concerns from this reviewer about non-linear MR methods, which is still evolving. Given this and the fact that the primary focus of this manuscript is not on exploring the shape of the associations, we have decided to remove the non-linear MR analyses from the manuscript, which was also requested by the Reviewer 3.

2. It is not clear why the authors did not adjust for physical activity when they examined the associations of adiposity measures with protein biomarkers.

Response: As suggested, we have now incorporated additional sensitivity analyses with additional adjustments for physical activity and other factors. As detailed in the manuscript, these adjustments did not significantly alter the observed associations (**eTable 10**; page 10, lines 193-195; page 15, lines 314-317).

3. To evaluate the causal effects of proteins on adiposity measures, the authors performed two-sample MR using GWAS summary statistics from Asian population. However, the GSs of adiposity measures were derived using IVs obtained from the trans-ancestry GWAS. I suggest adding sensitivity analyses using IVs from Asian population only.

Response: As suggested, we have conducted additional analyses using ancestry-specific genetic scores for BMI derived separately from the China Kadoorie Biobank (CKB) and the UK Biobank (UKB). Overall, approximately 75% and 12% of the trans-ancestry genetic score findings in CKB and UKB overlap with those obtained using CKB- and UKB-specific scores, respectively. However, the CKB-specific GWAS identified about 100 additional associations, which could indicate "overfitting" and/or "winner's curse" where the genetic score aligns too well with the original study dataset, limiting generalizability of the study findings. The lack of available East Asian-specific GWAS data for traits like BF%, and WHR complicates the use of purely East Asian instruments. Therefore, we opted to use trans-ancestry scores for broader applicability and also expanded the discussion on its potential limitations in the revised manuscript (**eFigure 9**; page 9, lines 171-174; page 16, lines 343-345).

4. Two proteins (BMPER and SCAMP3) showed bi-directional associations. Does this mean that the genetically predicted levels of BMPER and SCAMP3 have effects on adiposity measures and the genetic predicted adiposity measures also have effects on the expression of BMPER and SCAMP3? How to explain such phenomenon?

Response: Such bi-directional effects, as correctly interpreted by this reviewer, may suggest potential feedback mechanisms or shared genetic pathways between these proteins and adiposity, although the exact nature of these relationships requires further investigation. We have now added further discussion on the bi-directional associations identified for these two proteins (page 22, lines: 483-490).

5. Minor comment: Line 162 “The GSs of for BMI”.

Response: Now corrected (page 9, line: 165).

REVIEWER #2

1. The authors did not provide detailed information on how the genetic scores for body mass index, body fat percentage, waist circumference and waist-to-hip ratio were constructed. Detailed descriptions would help readers better understand the methodology and evaluate the validity of the genetic scores used in the analysis.

Response: As requested, we have now included a supplementary methods section that provides essential details on how the genetic scores were derived (Supplementary material, pages 4-5). A more detailed description of the method is available in a pre-print, which has been cited in the manuscript (reference 27).

2. In the Protein-to-Adiposity section, the authors did not describe the screening process for cis-pQTLs in detail. The corresponding pQTL information or complete GWAS data should be provided.

Response: As requested, we have now expanded the Methods section (page 9, lines 176-180) on GWAS methodology used for identifying *cis*-pQTLs. For clarification, further comprehensive details were provided in a preprint publication cited in the manuscript (reference 28), which includes the complete screening process. This preprint, initially focused on OLINK batch 1, now includes data from both batch 1 and batch 2 in its latest version (updated version will soon be made available in medRxiv), which is currently under revision.

3. In Figure 1a, the number of FDR sign proteins in the discovery set does not match the results section and eTable 7 of the manuscript. Please clarify the discrepancy.

Response: For clarification, the numbers of significant proteins shown in Figure 1a for the conventional analyses represent those that overlap between the conventional and genetic analyses, whereas the numbers in **eTable 7** correspond to any significant association found in the conventional analyses. We have now added a footnote to **Figure 1** to further clarify these.

4. In the colocalization analysis, the authors defined a posterior probability of $H_4 > 0.5$ as "strong evidence." However, it is more common to define a posterior probability of $H_4 > 0.8$ as "strong evidence." Please justify the chosen threshold.

Response: We have now updated the results section to include information (**eTable 13**) on the number of proteins that colocalise at different thresholds, including those with $H_4 > 0.8$ (page 19, lines: 395-399). Moreover, as suggested by this reviewer, we have revised the text accordingly, defining only those with $H_4 > 0.8$ as "strong evidence" of colocalisation.

5. In lines 425-428, the authors stated that these proteins with causal relationships with adiposity also have causal relationships with some cardio-metabolic diseases or their risk factors. They then directly inferred that adiposity might mediate the relationship between the proteins and cardiovascular metabolic diseases. This conclusion is not sufficiently rigorous. The authors should supplement the appropriate mediation analysis to support this viewpoint.

Response: While it is beyond the scope of this manuscript, we have conducted new mediation analyses for proteins demonstrating strong evidence of colocalisation ($H_4 > 0.8$). The findings from these analyses are discussed in the Discussion section (page 23, lines 500-505). A future paper is now planned that will further examine and clarify these associations.

6. I appreciate the authors' use of four indicators to describe adiposity from different perspectives and their classification of general adiposity and central adiposity. Unfortunately, the authors did not compare and discuss the proteins associated with these two different types of adiposity in the discussion section.

Response: Throughout the manuscript, we have, where appropriate, discussed and compared the associations of general and central adiposity with proteins (e.g., page 14, lines: 293-297; page 15, lines: 320-324; pages 17-18, lines 371-374, page 21, lines 456-458; page 22, lines 473-477). However, our ability to compare the proteins-to-adiposity genetic associations is constrained by the limited availability of published GWAS summary statistics for various adiposity traits from East Asian studies. This was further highlighted as a study limitation in the discussion of the manuscript (page 24, line 535-541).

REVIEWER #3

7. It is unclear to me why the authors performed adiposity-to-protein association testing using genetic scores. Genetic scores for adiposity traits can be subject to high horizontal pleiotropy as many instruments are pleiotropic. What efforts have the authors made to address risk of horizontal pleiotropy?

Response: We used genetic scores to explore the likely causal relationship between adiposity and levels of protein. Recognising the potential for horizontal pleiotropy, we

have now included sensitivity analyses using methods like IVW, MR-Egger, weighted median and MR-PRESSO to assess and account for potential pleiotropy (**eTable 12**). After thorough analysis, only 2% of proteins associated with BMI exhibited evidence of horizontal pleiotropy. However, after adjusting for multiple testing (FDR<0.05), none of the proteins showed evidence of horizontal pleiotropy. This suggests that the majority of the associations were robust and not significantly influenced by pleiotropic effects, supporting the validity of our findings (page 11, lines: 207-209; page 16, lines 345-347).

1. Following point #1, why didn't the authors use large-scale East Asian specific GWAS for adiposity traits to perform formal MR analyses? The authors used genetic scores with instruments' coefficients "weighted according to their effect sizes in UKB" - This can lead to bias due to population specificity. Overall, the adiposity-to-protein "causal relationships" are overstated.

Response: Done. Please see our response to Reviewer 1's third comment.

2. The four adiposity traits are strongly correlated with each other - I suggest adding multivariable MR (and perhaps multivariable regression in observational analyses as well) to quantify joint/conditionally independent effects of these traits on protein levels.

Response: Done. As suggested, we have now conducted additional analyses to quantify the conditionally independent effects of the four adiposity traits on protein levels, both for conventional and genetic associations. These analyses showed that, in conventional analyses, BMI, BF%, WC, and WHR independently affect the levels of 339, 1018, 26, and 620 proteins, respectively. Likewise, in the genetic analyses, we found that the levels of 9, 2, 1, and 90 proteins are affected by BMI, BF%, WC, and WHR, respectively, independent of each other. We have now updated the Methods (page 10, lines 191-195; page 11, lines 220-221), Results (page 14, lines 290-292; page 16, lines 336-339; **eFigure 5**), and Discussion (pages 21-22, lines 462-473) sections to cover these analyses and findings.

3. The study focused on consensus, but for associations that were different between Chinese and European population, do they represent population-specific biological mechanisms? I believe such associations are more interesting than those that could be replicated - isn't this the point of having resources for ancestrally diverse populations?

Response: We thank this reviewer for highlighting the importance of exploring population-specific biological mechanisms. While we agree that such differences could reveal interesting insights and underscore the value of resources for ancestrally diverse populations, the primary focus of this current manuscript is to clarify the associations within East Asian populations and to replicate key findings in Western populations. Exploring the distinct associations between Chinese and European populations represents an important area of future research, and we plan to investigate these population-specific mechanisms in subsequent manuscripts.

4. In protein-to-adiposity MR analyses, again, how did the authors address

risk of horizontal pleiotropy? Can a cis-pQTL be eQTL or sQTL of another gene that does not code for the assayed protein? In general, the MR analyses in the current form are overly simplistic and have few sensitivity analyses. The authors should consider accompanying the analyses with the STROBE-MR checklist.

Response: We implemented these analyses using *cis*-pQTLs, as they are less prone to horizontal pleiotropy. Among the 10 proteins showing both MR and colocalization evidence of causal links, none of the *cis*-pQTLs used were trans-pQTLs for other proteins. We plan a separate report to combine all pQTLs due to the large number of trans-pQTLs and eQTLs identified and their more complex horizontal pleiotropic effects compared to *cis*-pQTLs. This has now been added to the discussion as part of the study's limitations and implications for future research (page 24, lines 541-543).

5. For colocalized associations, please add locus-zoom plots for visual inspection.

Response: Done (see eFigure 12).

6. Some of the proteins showing significant associations seem to be in the MHC region. These should not be the focus of interpretation due to exceedingly high risk of pleiotropy and complicated LD structure.

Response: We thank this reviewer for pointing out the involvement of proteins in the MHC region, which is known for its complex LD structure and high risk of pleiotropy. Among the 31 proteins showing causal relevance for adiposity levels, only 1 protein (CFB) is in the MHC region (chr6:28,477,797-33,448,354). We have now revised the manuscript to emphasise that the findings related to CFB should be interpreted with caution. Additionally, we have provided further discussion on the implications of pleiotropy and LD in the MHC region to ensure a balanced and informed interpretation of these findings (page 22, lines: 490-493).

7. Non-linear MR, especially the adopted residual stratification method, has been criticized. The revised doubly-ranked stratification method has also been shown to introduce bias. I suggest removing the non-linear MR analyses.

Response: Done. Please see our response to the Reviewer 1's first comment.

8. The enrichment test did not seem to account for the background. Are the Olink-measured ~3,000 proteins already enriched in cardiometabolic pathways?

Response: We acknowledge the potential pre-enrichment of certain proteins, especially those related to cardiometabolic pathways in the Olink Explore panel. To address this, we have now included a sensitivity analysis to account for the background enrichment of these pathways, in which we compared all 2923 OLINK proteins to all proteins with annotations. In this analysis, we found that none term was enriched for BMI, one term was enriched each for WC and WHR, while again, 3

lipids metabolism related terms were enriched for WHR (page 11, lines 213-215; page 18, lines 375-379; **eTable 14**).

9. Will the pQTL summary statistics be openly available?

Response: A major GWAS paper on the ~3000 proteins is currently under review, and once published, the pQTL summary statistics will be made available openly.

REVIEWERS' COMMENTS:

Reviewer #1 (Remarks to the Author):

The authors have addressed all my concerns.

Reviewer #3 (Remarks to the Author):

The authors have addressed my comments.